# The Gut Microbiome Alterations in Pediatric Patients with Functional Abdominal Pain Disorders

**DOI:** 10.3390/microorganisms9112354

**Published:** 2021-11-15

**Authors:** Bassam Abomoelak, Veronica Pemberton, Chirajyoti Deb, Stephani Campion, Michelle Vinson, Jennifer Mauck, Joseph Manipadam, Sailendharan Sudakaran, Samit Patel, Miguel Saps, Hesham A. El Enshasy, Theodoros Varzakas, Devendra I. Mehta

**Affiliations:** 1Pediatric Specialty Laboratory, Arnold Palmer Hospital, Orlando Health, Orlando, FL 32806, USA; vfnderrick@gmail.com (V.P.); chirajyoti.deb@orlandohealth.com (C.D.); Stephani.campion@orlandohealth.com (S.C.); Joseph.manipadam@orlandohealth.com (J.M.); samit.patel@orlandohealth.com (S.P.); 2College of Medicine, Florida State University, Tallahassee, FL 32306, USA; michelle.vinson@globalaes.com (M.V.); Jenn.mauck@gmail.com (J.M.); 3Wisconsin Institute for Discovery, University of Wisconsin, Madison, WI 53706, USA; sudakaran@wisc.edu; 4Pediatric Gastroenterology, Hepatology and Nutrition Division, University of Miami Miller School of Medicine, Miami, FL 33136, USA; msaps@med.miami.edu; 5Institute of Bioproduct Development (IBD), Universiti Teknologi Malaysia (UTM), Skudai 81310, Johor, Malaysia; henshasy@ibd.utm.my; 6School of Chemical and Energy Engineering, Faculty of Engineering, Universiti Teknologi Malaysia (UTM), Skudai 81310, Johor, Malaysia; 7City of Scientific Research and Technology Applications (SRTA), New Burg Al Arab, Alexandria 21934, Egypt; 8Food Science and Technology, University of Peloponnese, 24100 Kalamata, Greece; t.varzakas@uop.gr

**Keywords:** gut microbiome, FAPDs, dysbiosis

## Abstract

In this prospective longitudinal study, we enrolled 54 healthy pediatric controls and 28 functional abdominal pain disorders (FAPDs) pediatric patients (mean age was 11 ± 2.58 years old). Fecal samples and symptom questionnaires were obtained from all participants over the course of the year. Clinical data assessment showed that FAPDs patients were more symptomatic than the control group. Microbiome analysis revealed that Phylum Bacteroidetes was higher in FAPDs compared to the control group (*p* < 0.05), while phylum Firmicutes was lower in FAPDs (*p* < 0.05). In addition, Verrucomicrobiota was higher in the control group than the FAPDs (*p* < 0.05). At the genus level the relative abundance of 72 bacterial taxa showed statistically significant differences between the two groups and at the school term levels. In the control group, Shannon diversity, Observed_species, and Simpson were higher than the FAPDs (*p* < 0.05), and beta diversity showed differences between the two groups (PERMANOVA = 2.38; *p* = 0.002) as well. Using linear discriminant analysis effect size (LEfSe), *Enterobacteriaceae* family and *Megaspherae* showed increased abundances in vacation term (LDA score > 2.0, LEfSe, *p* < 0.05). In the FAPDs group, the severity of symptoms (T-scores) correlated with 11 different taxa bacterial relative abundances using Pearson′s correlation and linear regression analyses. Our data showed that gut microbiome is altered in FAPDs compared to the control. Differences in other metrics such as alpha- and beta diversity were also reported between the two groups. Correlation of the severity of the disease (T-scores) correlated with gut microbiome. Finally, our findings support the use of *Faecalibacterium/Bacteroides* ratio as a potential diagnostic biomarker for FAPDs.

## 1. Introduction

Approximately 32% of schoolchildren experience weekly abdominal pain with 20% of them reporting functional disability including school absenteeism, and limitation with social and physical activities [1]. In 1958, Apley et al. described a group of children with chronic abdominal pain and introduced the term recurrent abdominal pain (RAP) [2]. It was later recognized that this was a “waste basket” term, as it included children with abdominal pain with and without an organic etiology. In 1999, the first pediatric version of the Rome II criteria was published. The Rome committee recognized the shortcomings of Apley′s definition and replaced the old term of RAP. Since then, there have been three iterations of the pediatric Rome criteria [3] with the latest version (Rome IV) defining children with abdominal pain lasting 2 or more months in the absence of an organic cause to their symptoms as, functional abdominal pain disorders (FAPDs) [4]. FAPDs as a group, include four distinct diagnoses, functional dyspepsia, irritable bowel syndrome (IBS), functional abdominal pain and NOS (not otherwise specified), and abdominal migraine. The prevalence of the different FAPDs varies worldwide with studies showing the prevalence of IBS ranges from 1.2% to more than 21% [5].

Despite the high prevalence and impact of FAPDs in children, their etiology and pathogenesis remain poorly understood. In search of risk factors that could worsen or trigger these disorders, investigators have found that in children, abdominal pain was more common and severe during winter months, and this seasonal pattern is not found in adults [1,6,7,8]. Multiple factors can explain this winter predominance. Some studies have shown that in areas of cold weather (not as clearly reported in areas of temperate climate) children experience more abdominal pain during winter, while other investigators proposed a role of school stress, sleep, or diet to explain the seasonal predominance of symptoms. The role of each of these factors or the mediators contributing to the symptoms are yet to be identified. To explore this further, we reasoned that the geographical location of the conducted study, Orlando region, offered us a chance to separate the seasons from school semesters. Florida is categorized as a subtropical state (humid subtropical and tropical wet-and-dry). In the year of the study (2017), the temperature index for the Orlando region showed maximum temperatures in June and July (high 92F, low 75F), while November, December, and January months showed an average of high 76F and low 56F (data not shown). The gut harbors more than 500 species of bacteria that play a role in host nutrient acquisition, modulation of host gene expression, and regulation of host immune system [9,10,11]. In healthy individuals, the gut microflora composition is balanced, but the balance can be disturbed in case of disease. Recent studies have suggested a possible role of microbiota in the pathophysiology of multiple gastrointestinal and extraintestinal diseases including IBS, inflammatory bowel disease (IBD) [12,13,14,15], obesity, and Type 1 and 2 diabetes [16]. Several studies have also linked bacterial genera abundance to GI diseases. For example, *Faecalibacterium* and *Dorea* showed altered abundances in the gut of IBS patients [17]. Similarly, in IBD patients, *Bacteroides* may be involved in the development of disease [14]. Microbiome alterations not only cause local changes, but also may result in central nervous system effects which may explain some of the extraintestinal symptoms reported by children and adults [18].

We conducted a study to characterize the microbiota and their changes in children with and without FAPDs throughout the year in a subtropical region devoid of four seasons. We hypothesized that the microbiota composition of children with and without FAPDs will differ, and there will be variations in microbiota when comparing school term versus vacation.

## 2. Materials and Methods

### 2.1. Participants and Study Design

This study was a prospective longitudinal study of children ages between 7 and 16 years of age with and without FAPDs. Cases (FAPDs patients) were recruited at a large pediatric gastroenterology clinic in Central Florida (Center for Digestive Health and Nutrition at Arnold Palmer Hospital for Children, Orlando, FL, USA), while the controls (no FAPDs) were enrolled from five local pediatric practices. Enrollment was conducted by clinic staff. Participants were consented via child and/or parental consent. Research forms were managed using REDCap electronic data capture tools hosted at Florida State University (FSU, Tallahassee, FL, USA) [19]. The project was approved by FSU Institutional Review Board (IRB, Tallahassee, FL, USA) ethics committee (One Florida, Study ID: IRB 201701009).

### 2.2. Inclusion and Exclusion Criteria

An experienced gastroenterologist from Orlando Health performed Rome IV classification. The Rome IV Diagnostic Questionnaire on Pediatric Functional Gastrointestinal Disorders (for parents and children) was used to ensure FAPD patients satisfied Rome IV criteria for inclusion (Appendix A). Patients were excluded if they had an acute illness (e.g., appendicitis, gastroenteritis, pneumonia, etc.) or a chronic disease, such as cancer, inflammatory bowel disease (IBD), celiac disease, stomach ulcer, current febrile illness, or were actively taking antibiotics or had completed a course of antibiotics two weeks prior to enrollment.

### 2.3. Data Collection

Throughout the year, FAPD patient′s clinical data, use of antibiotics or probiotics as well as stress data was collected through well-defined and validated stress and symptoms scales. Initial clinical assessments were completed at enrollment and additional data were submitted to the investigators via US mail. The stress score was evaluated using PROMIS^®^ Stress Score. The PROMIS^®^ is standardized to generate T-scores [20]. The symptom intensity form assessed eight of the most common pediatric gastroenterology symptoms in children. Patients (or their caregiver) were asked to “rate each gut problem” from 0 (“does not bother me at all”) to 10 (“the most bothersome problem I can imagine”); higher number of symptoms is indicative of more clinical concern (Appendix A). The symptoms score is not standardized but it captures key symptoms and severity.

### 2.4. Stool Samples Collection

Patients and caregivers were instructed to collect three stool samples at home and send them via US mail. Every FAPDs patient and healthy control gave three stool samples throughout the year. The hemoccult ICT collection kit was used to collect all stool samples (Beckman Coulter, Brea, CA, USA). The kit included a toilet hat, application stick, collection card, collection pouch, specimen biohazard bag, pre-addressed mailing envelope, and detailed paper instructions about the collection procedures. The procedure has been validated as a cost-effective method of stool collection with no significant microbiome differences from other stool collection methods [21]. Stool samples were frozen at −20 °C until further use.

### 2.5. Data Management

For analyses, data was grouped into academic terms as school term (August–May) and vacation term (June–August). Changes in sample size are related to missing data issues (i.e., patient did not submit a sample) or from the decisions on data management as described above.

### 2.6. Statistical Analyses

(*I*) Clinical outcomes. Patient characteristics are reported as mean ± SD or count (%). The independent sample t-test, and the Fisher′s exact test were used to test group equivalence on baseline characteristics. Symptoms were grouped into two groups: symptomatic (1–10) and asymptomatic (0) and reported as frequency with percentages. Fisher′s exact test was used to assess the association of symptoms and study group (FAPDs and controls). Among FAPDs patients, gut symptom intensity by academic term (school vs. vacation) was assessed with the Friedman′s test and the Wilcoxon signed-rank test, respectively. Tests were two-tailed and α = 0.05 was used for statistical significance. IBM SPSS Statistics for Windows v. 26.0 was used for univariate analyses. Psychological stress was collected only for the FAPDs group using validated assessment forms. The Patient-Reported Outcomes Measurement Information System PROMIS^R^ was used for self-assessment (if the child was >7 years old) or the parent form (if the child age was <7 years old). The raw scores were transformed into T-scores where the increase of T-scores means increase of disease severity. A score of 50 represents the average psychological stress level for normal children [22].

*(II)* Bacterial relative abundance. Multivariate analysis by linear models (MaASLin) was used to assess differences in relative abundance of bacterial phyla between groups (FAPDs vs. controls), and for academic year (school vs. vacation). GraphPad Prism V.8 was also used for the analyses; two-tailed, and a *p* < 0.05 was used for statistical significance. Associations of specific bacterial taxa with other variables such as school or vacation terms were assessed using the linear discriminant analysis effect size (LEfSe). (MaAsLin) and LEfSe statistical pipelines from Huttenhower lab Galaxy (http://huttenhower.sph.harvard.edu/galaxy, accessed on 8 November 2021) were used in the analysis of the data.

*(III)* Correlation of the severity of the disease with the relative abundances of bacteria in FAPDs patients. IBM SPSS Statistics for Windows v. 26.0 was used to conduct Pearson correlation of the T-scores with the relative abundances of bacteria taxa in FAPDs patients. Linear regression was then used to measure the correlation between the identified taxa. Only FAPDs patients with a T-score were included in this analysis. Any patient with a missing T-score was excluded when conducting Pearson′s correlation. Outliers for each bacterial taxa were identified in SPSS by using a boxplot. Values denoted by the * symbol are called extreme values within SPSS and were removed for each bacterial taxa before conducting the correlation. Additionally, bacterial taxa that had zeroes for more than 70% of the patients were also excluded when conducting the Pearson′s correlation. The receiver operating characteristics curve (ROC) was performed using SPSS as well.

### 2.7. DNA Extraction, Next Generation of Sequencing, and Bioinformatics Analysis

Total genomic DNA was extracted from the collected stool samples using QIAGEN kits (QIAGEN, Los Angeles, CA, USA). The DNA was analyzed for purity and integrity using Agilent Bioanalyzer (Agilent, Santa Clara, CA, USA), and quantified by labeling and detection kits (Invitrogen, Waltham, MA, USA). DNA was used to amplify the V3–V4 region of the bacterial 16S rRNA gene. Microbiome analysis was performed by the UW biotechnology center using Quantitative Insights Into Microbial Ecology (QIIME2) version 2 [23]. Illumina sequencing reads were denoised and quality filtered using the denoising program DADA2 (Appendix A). This step trimmed low quality bases, filtered out noisy sequences, corrected errors in marginal sequences, removed chimeric sequences and singletons, joined denoised paired end reads, and then dereplicated those sequences. The resultant dereplicated sequences were termed as “Amplicon sequence variant (ASV)”. Sequence variants were aligned and masked using Mafft and the phylogenetic tree of the ASV′s created using FastTree. Taxonomy was assigned using Bayesian classifier based on a pretrained Silva database curated to the exact 16s amplicon region. The resulting biome formatted table describing the occurrence of bacterial phylotypes within each sample was generated for further downstream analysis. Low frequency reads (<0.01%) were filtered from the Biome-formatted table. Alpha rarefaction curves using Shannon, Simpson, and Observed-species were calculated for all samples with a rarefaction upper limit of median depth/sample count and the alpha diversity between different treatments were compared using Wilcoxon signed-rank test. Samples were removed from further characterization if they did not contain sufficient reads. Beta diversity was calculated, and ordination plots were generated using Bray–Curtis and Jaccard (Non-Phylogenetic), weighted and unweighted Unifrac (Phylogenetic) on ASV data leveled according to the lowest sample depth.

## 3. Results

### 3.1. Patient Characteristics

Eighty-two patients were included in the analyses (controls *n* = 54, and FAPDs *n* = 28). Overall, mean age was 11 ± 2.58 years old. Just over half of the children were female (43/82; 52%) and non-Hispanic (50; 61%). The groups were similar in age (*p* = 0.287), sex (*p* = 0.64), and race/ethnicity (*p* = 0.400). During enrollment, there were no statistically significant differences in antibiotic course among FAPDs and control groups (*p* = 0.086). More specifically, 53 (65%) children had not taken antibiotics in the past year. Among the 28 (35%) children that had taken antibiotics over the past year, 57% (16 patients) had received one course and 43% (12 patients) had received two or more courses. FAPD children were significantly more likely to be on probiotics (10 vs. 1, FAPDs and controls, respectively, *p* < 0.001) and among children taking probiotics (6 participants (7%) reported taking daily dosages). During the duration of the study, FAPDs patients reported significantly higher symptom intensity than healthy controls (Appendix A). Briefly, the patients were classified as follows: IBS *n* = 15, functional dyspepsia *n* = 13, abdominal migraine *n* = 7, functional constipation *n* = 5, adolescent rumination syndrome *n* = 2, functional abdominal pain (NOS or non-otherwise specified) *n* = 1, functional nausea *n* = 1, functional vomiting *n* = 1, and cyclic vomiting syndrome *n* = 1. No patient belonged to non-retentive fecal incontinence or Aerophagia groups. In addition, the 13 patients with functional dyspepsia were classified as either post-prandial distress syndrome *n* = 10 (77%) or epigastric pain syndrome *n* = 3 (33%).

### 3.2. Bacterial Relative Abundances between the Control and the FAPDs Groups

In total, 213 stool samples were sequenced including 142 control samples and 71 FAPDs. A total of 4,156,326 sequences were generated and 7214 ASVs were identified in our analysis. MaASLin analysis revealed differences between the two groups at the phyla levels (Appendix A). Briefly, the predominant phyla in control and FAPDs groups were Bacteroidetes, Firmicutes, Proteobacteria, Actinobacteria, and Verrucomicrobiota. Figure 1 depicts the relative abundances of the predominant phyla in the two groups. While Bacteroidetes were lower in the control group compared to the FAPDs group (*p* < 0.05), Firmicutes were higher in the FAPDs group than the control group (*p* < 0.05). Similarly, Verrucomicrobiota were higher in the control group compared to the FAPDs group (*p* < 0.05). On the other hand, the Actinobacteria and Proteobacteria did not show statistically significant differences between the two groups (all *p*_s_ > 0.05).

At the genus levels, 38 bacterial taxa showed statistically significant differences between the two groups, and 34 bacterial taxa between the two school terms. Interestingly, *Bacteroides* relative abundances were affected in FAPDs and in school vacation. With the exception of *Bacteroides*, *Eubacterium*, *Candidatus-stoquefishus*, *Clostridium-innocuum group*, *Anaerofustis*, *Eisenbergiella*, *Acetanaerobacterium*, and *Eggerthella* all the remaining 30 species were depleted in the FAPDs group compared to the control group (MaAsLin, *p* < 0.05). *Faecalibacterium* species and *Ruminococcus* species were depleted in the FAPDs group compared to the controls (*p* < 0.001). Other bacterial taxa such as *Sutterella*, *Akkermansia*, *Dialister*, and *Veillonella* were depleted from the FAPDs groups (*p* < 0.05) as well. Figure 2 and Figure 3 depict the major bacterial taxa differences between the FAPDs and the control groups. Per school term, only *Bacteroides* and *Holdemania* were depleted in the vacation term compared to the school term, while the other 32 were enriched in the vacation term. All the identified taxa are included in the file (Appendix A). To investigate whether bacterial taxa differ per school season, we used LEfSe software to perform the analysis. The samples were analyzed using the group (control or FAPDs) as a class, while the school terms (school or vacation) were a subclass. *Enterobacteriaceae* and *Megasphaera* showed statistically significant differences in the vacation term (LDA score > 2.0, *p* < 0.05, LEfSe) (Appendix A).

### 3.3. Microbial Diversity between the Control and the FAPDs Groups

Alpha diversity analyses were conducted for the control and the FAPDs groups. Shannon, Simpson, and Observed species indices showed that the two groups differ in alpha diversity. The three indexes were higher in the control group than the FAPDs group (Figure 4) (Mann–Whitney, *p* < 0.05). Beta diversity analysis using Bray–Curtis dissimilarity index confirmed the structure differences of fecal microbiota between the FAPDs and the control groups (permutational multivariate analysis of variance (PERMANOVA) = 2.38; *p* = 0.002) (Figure 5).

### 3.4. Correlation of Bacterial Relative Abundances and the Severity of the Stress in FAPDs Group

FAPDs patients received a T-score (Appendix A). Any patient with missing T-score was removed from the analysis and outliers were excluded from the analysis as well. A total of 65 T-scores were used in the correlation. To correlate the severity of the disease with the relative abundances of some bacterial taxa, we used Pearson correlation. In total, 11 bacterial taxa were found to correlate significantly with the T-scores (*p* < 0.05). Two bacterial taxa were found to be positively correlating with the T-score, while the remaining nine taxa were found to be negatively correlating with the T-score. *Blautia* and *Colidexbacter* correlated positively with the T-scores. We conducted linear regression to measure the correlation (Figure 6). The other bacterial taxa showing negative correlation are *Alistipes*, *Parabacteroides*, *Fusicatenibacter*, *Lachnospiraceae-NK4A136-Group*, *[Eubacterium]-Eigens-Group*, *[Eubacterium]-Ventriosum-Group*, *Monoglobus*, *UCG-002*, and *Akkermansia*.

### 3.5. Faecalibacterium Versus Bacteroides (F/B) Ratio as a Potential Diagnosis for FAPDs

Post hoc analysis was performed based on observations of the inverse relationship between *Faecalibacterium* and *Bacteroides* in the FAPDs and control groups. To validate the F/B ratio in FAPDs diagnosis, we calculated the individual ratios for the two groups after excluding the outliers. For the control group (*n* = 131), the mean was 0.20 (±0.12), while the mean for the FAPDs group (*n* = 63) was 0.13 (±0.10). We conducted ROC analysis and the area under the curve (AUC) was 0.712. When we performed the analysis for the school term to exclude vacation impact, the AUC increased to 0.797. The ROC data for both analyses are in the Appendix A.

## 4. Discussion

FAPDs are a heterogenous group that includes IBS, functional dyspepsia, abdominal migraine, and functional abdominal pain—not otherwise specified. Previously, in a retrospective cohort study collected from six US tertiary care institutions, the rates of abdominal pain consultations were consistently higher in winter in all the six locations [6,7]. This study raised the question of whether this phenomenon was seasonal, or school term related. As Florida is categorized as a subtropical state (humid subtropical and tropical wet-and-dry) devoid of seasons, we were able to selectively assess the impact of school terms and vacation. Our study is a prospective longitudinal study, with patients who met Rome IV criteria for FAPDs and were compared with healthy controls. FAPDs patients were more symptomatic than the control group, especially during school terms, and as expected, abdominal pain was significantly higher (*p* < 0.001).

At the microbiome level, our findings confirmed that the two groups (control and FAPDs) exhibited differences at the phyla and genus levels, as well as α-diversity and β-diversity. Approximately, 12% of the identified bacterial taxa (38 out of 308 total) showed significant differences between the two groups which is a clear indication of microbiome signature differences between FAPDs and the healthy controls. Bacteroidetes and Firmicutes were significantly different between groups: Bacteroidetes was lower in the control group compared to the FAPDs (Mann–Whitney, *p* < 0.05), while the *Firmicutes* were higher in the control group (Mann–Whitney, *p* < 0.05). Our findings are consistent with some broader trends reported in a metanalysis of IBS in adults, suggesting that the abundance of Firmicutes is beneficial and *Bacteriodetes* can be detrimental [24]. In addition, our data showed that only *Enterobacteriaceae*, and *Megaspherae* showed school and vacation term variations (LEfSe, *p* < 0.05). This finding implies a minor role of school stress in microbiome alterations on the FAPDs patients. Our study controlled for the recent antibiotics consumption as an exclusion criteria from the study, but other factors such as diet and pre- and probiotics consumption were not controlled due to the nature of recruitment of pediatric population. These factors represent a limitation of our study. Unpublished data suggested a minor role of the probiotics consumption on the FAPDs gut microbiome in our study (data not shown).

Interestingly, T-scores were found to correlate with 11 bacterial taxa in the FAPDs group. *Blautia* was among the two bacterial taxa that were positively correlated with the severity of the symptoms. *Blautia* depletion was found to be associated with childhood obesity and intestinal inflammation [25]. The role of *Blautia* in FAPDs severity should be further investigated. The remaining nine bacteria taxa with a negative correlation with T-scores included *Akkermansia* which is a beneficial bacterium in glucose metabolism, lipid metabolism, and intestinal immunity [26,27]. *Akkermansia municiphila* is also considered a potential probiotic bacterium [28]. Interestingly, the T-scores did not correlate with the major gut bacterial taxa and among the 38 differential bacterial taxa in FAPDs compared to the control group, five bacterial taxa were among the 11 taxa correlated with T-scores. The clear distinction between the microbiome of the FAPDs and the healthy control raises an interesting question about the possible role of bacterial metabolites in the FAPDs etiology, especially with approximately 12% of the total gut microbiome affected. The exact role of the 11 bacterial taxa in the severity of the symptoms remains to be elucidated, especially with the potential role of such bacterial taxa as probiotics.

Interestingly, *Faecalibacterium* was depleted in FAPDs group, and, among others, this bacterium is involved in short chain fatty acids (SCFAs) (butyrate and acetate) as well as lactic acid production [29]. *Faecalibacterium*, *Clostridium*, and *Roseburia* are the main producers of butyrate, and they belong to the phylum Firmicutes, thought to be protective. *Veillonella* and *Akkermansia* (Verrucomicrobia) both produce propionic and acetic acids. Higher levels of organic acid producing flora, and especially acetic and propionic acid, were found to be associated with IBS with *Veillonella* and *Lactobacillus* specifically implicated [29]. The relationship between gut microbiome and IBS, however, is likely more complicated than just SCFAs production, and may include other metabolites, proteases, and other host–microbe interactions [30,31,32]. The identified bacterial taxa were found to be involved in energy production, inflammation, and gut homeostasis. Although their roles in the establishment of FAPDs symptoms are not fully understood, their roles in mucosal integrity and innate immunity need to be elucidated [33].

Our post hoc analysis showed *Faecalibacterium*/*Bacteroides* ratio alterations in the FAPDs group compared to the control group. Previous data of gut microbial dysbiosis focused on the ratio of Bacteroidetes and Firmicutes as a potential IBS biomarker [34,35], but in our study we showed the validity of the *Faecalibacterium/Bacteroides* ratio at the genus level.

## 5. Conclusions and Future Directions

Our findings revealed the gut microbiome differences at the phylum and genus levels between FAPDs pediatric patients and healthy controls. The F/B ratio suggests the potential use for FAPDs diagnosis. If confirmed in a larger study, this finding could be used as a future biomarker for FAPDs, disease progression, and intervention evaluation. The analysis of fecal metabolites and functional genomics will allow deeper insights into other cellular and molecular pathways for FAPDs diagnosis. To the best of our knowledge, our study is the first to report significant variation in FAPDs microbiome. Though limited by the sample size, our findings support larger studies assessing factors affecting microbiome changes the role of school in FAPDs.

## Figures and Tables

**Figure 1 microorganisms-09-02354-f001:**
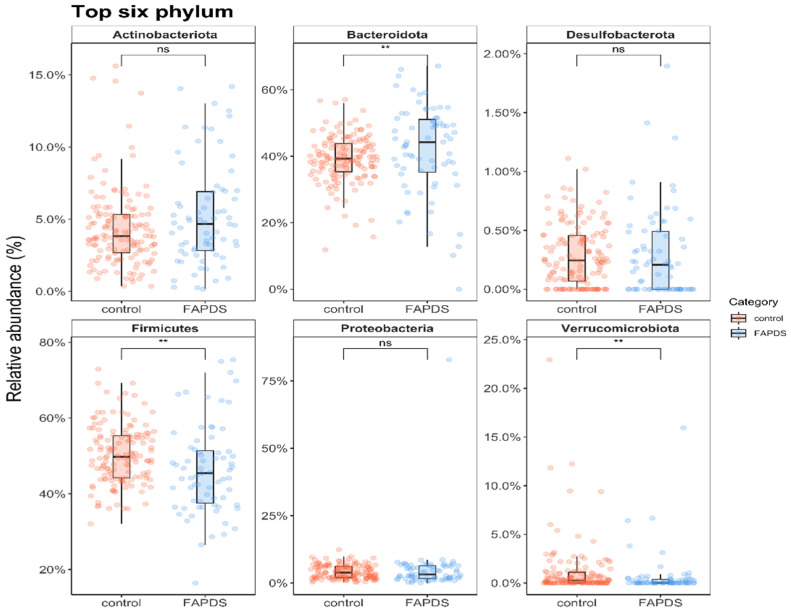
The relative abundances of the six identified phyla between the FAPDs and the control groups (*p*-value of <0.05 was considered significant). ** denotes significance, while ns refers to non-significant.

**Figure 2 microorganisms-09-02354-f002:**
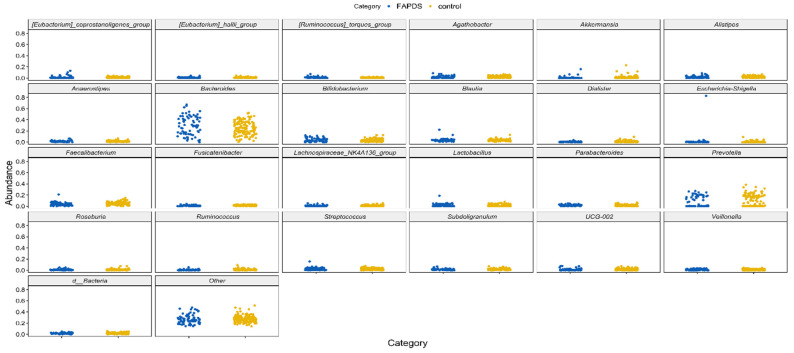
The relative abundances of the most abundant bacterial taxa in the study.

**Figure 3 microorganisms-09-02354-f003:**
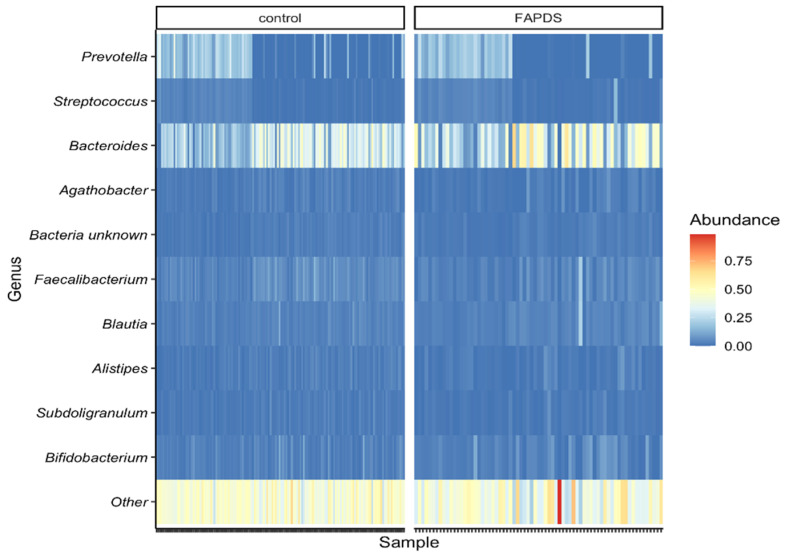
Heatmap illustrating the top ten identified bacterial taxa between the two groups.

**Figure 4 microorganisms-09-02354-f004:**
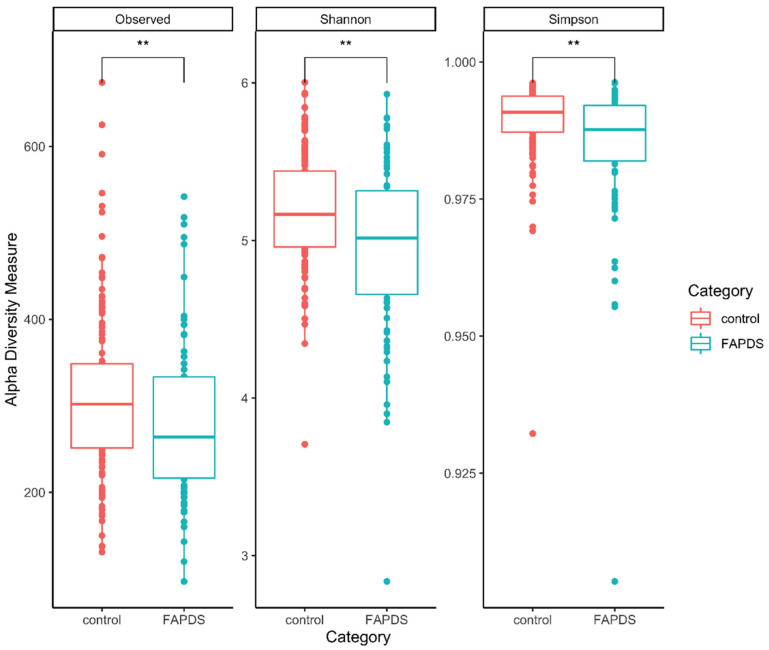
Alpha diversity comparison between the two groups. Shannon, Observed species, and Simpson were higher in the control group than the FAPDs group (Mann–Whitney, *p* < 0.05). ** denotes significance.

**Figure 5 microorganisms-09-02354-f005:**
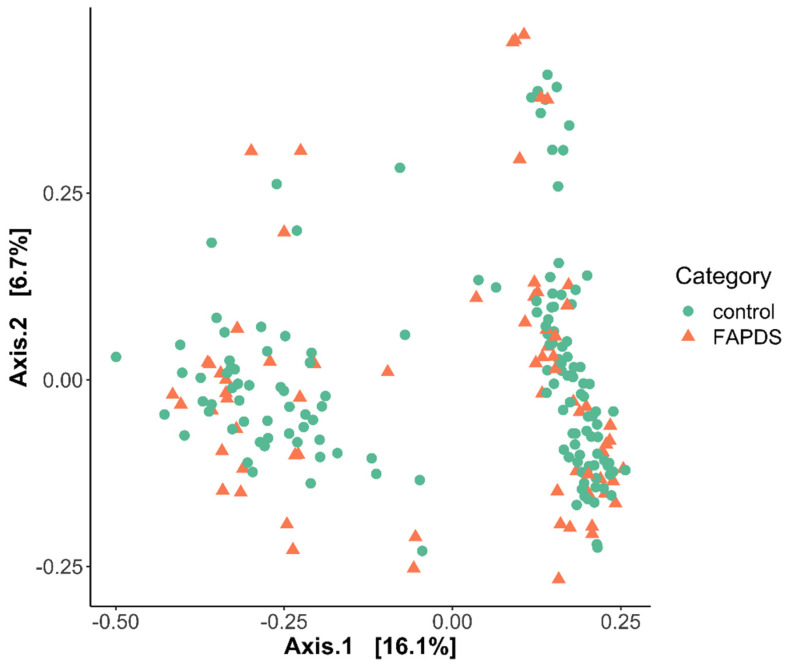
Principal component analysis of gut microbiota composition in the FAPDs and the control groups. Green circles represent the control, while red triangles denote the FAPDs group (PERMANOVA = 2.38; *p* = 0.002).

**Figure 6 microorganisms-09-02354-f006:**
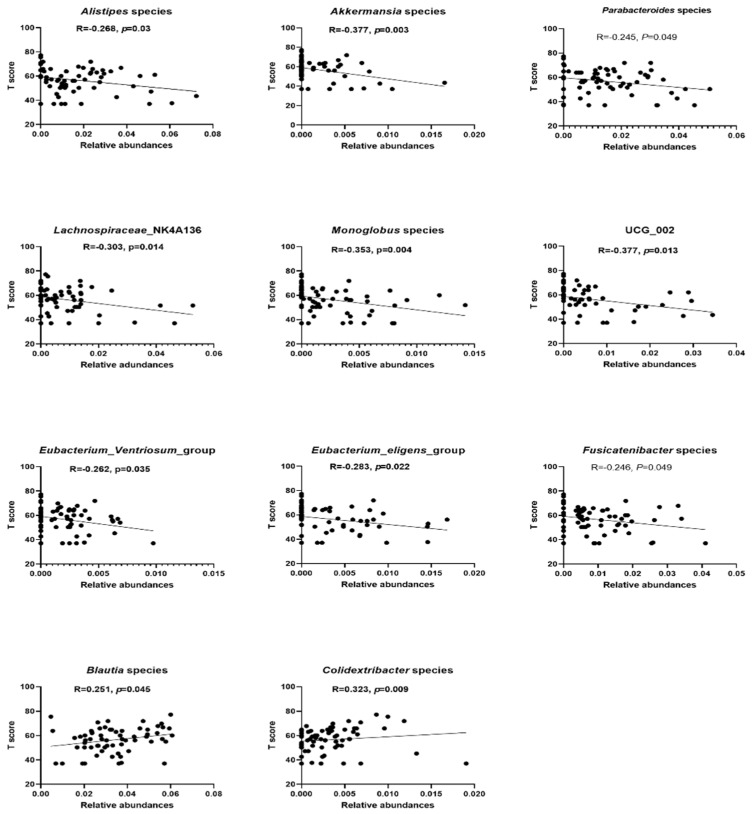
Correlation between the T−scores and the depleted and enriched stool microbiota taxa. A total of 11 bacterial taxa showed significant correlation by Pearson′s (London, UK) (*p* < 0.05). *Blautia* and *Colidextribacter* showed positive correlation with T−scores, while the remaining nine taxa showed negative correlation. R values and *p*-values for every taxa were included in the graph.

## Data Availability

The data will be available when the article is accepted for publication.

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
