# Peer review of "The Gut Microbiome Alterations in Pediatric Patients with Functional Abdominal Pain Disorders"

_microorganisms, 2021, doi:10.3390/microorganisms9112354_

Round 1

Reviewer 1 Report

The authors present the results of a prospective clinical study aiming to describe differences in gut microbiota composition between pediatric patients with functional abdominal pain disorders and healthy controls.

The analysis revealed distinct differences concerning gut microbiota composition in feces, including the prevalence of phyla Firmicutes and Bacteroidetes.

However, some major remarks, mainly on the study design, are necessary.

Major remarks

  • The exact study design is not clear: at what time points did participants collect stool samples? Were those time points comparable between individuals?
  • Did you reveal any differences in gut microbiota composition in between time points analysed?
  • How long did mailing of specimens take until specimens were frozen at -20°?
  • Did intake of probiotics have an impact on gut microbiota composition in patients with functional disorders? This is a major bias from my point of view (10/28 in FAPD group).
  • Did symptoms differ between individuals on probiotics and those without probiotics?
  • Did you assure that individuals were not on antibiotics in advance of every stool sample?
  • Could you add some discussion on the function of taxae that were depicted with significant differences between groups?

Minor remarks

  • Most authors use italic letters only for Genera and lower taxonomic levels - the use of italic letters also for phyla is a little confusing

Author Response

Reviewer 1

Comment 1. The exact study design is not clear: at what time points did participants collect stool samples? Were those time points comparable between individuals?

Reply. FAPDs conditions are thought to be seasonal in pediatric population and they are idiopathic in nature. The symptoms associated with FAPDs are defined in ROME IV criteria (criteria is attached in the supplementary files), and the clinical community settled on the seasonality of the symptoms. Seasonality means weather and school as the symptoms of FAPDs are worse in winter and school time. Although, the reason(s) behind this seasonality is not accurately identified, we suggested that gut microbiome alterations can cause these symptoms, especially other data showed that Irritable Bowel Syndrome (IBS) gut microbiome, which is subcategory of FAPDs, is different from healthy subjects. The novelty of our study is that we addressed FAPDs in pediatric population which is generally hard to recruit in clinical setting. The stool samples were collected three times from every FAPDs patient and every control participant throughout the year. As Florida has a mild weather all year long (no weather seasonality), we decided to compare gut microbiome in school (August to May) and summer vacation (June to August). Our study is focused on comparing the gut microbiome of FAPDs patients and healthy controls in two timepoints (School VS summer vacation). The collection time points were comparable among the patients and healthy controls as our research coordinator will follow up with them through phone calls to remind them to send the stool samples. Some participants came to the clinic to give the stool samples. Phone calls and follow up procedures were described in the IRB of the study.

Comment 2. Did you reveal any differences in gut microbiota composition in between time points analyzed?

Reply. We conducted the analysis between the time points for FAPDs patient stool samples using MaAsLin pipeline. The comparison was performed between FAPDs patients in school and summer vacation. None of the major bacterial taxa showed statistically significant differences between the two time points (p>0.05).  We concluded that FAPDs gut microbiome is statistically the same in both time points.

Comment 3. How long did mailing of specimens take until specimens were frozen at -20°?

Reply. All the participants of the study were given full kits for stool collection and instructions. Our research coordinator explained all the steps during the recruitment and consenting process. In addition, the kit included pre-stamped envelope with biohazard bags for mailing the samples. All the samples were sent to the lab the following day and they were frozen. Collection kit included hemoccult that was used for microbiome studies sample collection with no apparent discrepancy in results (Dominianni et al., 2014). We provided the reference supporting the hemoccult method for collection methods in the manuscript.

Comment 4. Did intake of probiotics have an impact on gut microbiota composition in patients with functional disorders? This is a major bias from my point of view (10/28 in FAPD group).

Reply. In some cases, probiotics can be part of the treatment for FAPDs cases, and we cannot ask the patients to stop probiotics to enroll in the study, but probiotics consumption data were collected from the patients as outlined by the IRB rules. We compared the gut microbiome in FAPDs for probiotics effect using MaAsLin, and only Alistipes and Roseburia showed statistically significant difference between the two cohorts (p<0.05). No additional relevant taxa were found to be significantly different between the two cohorts (p>0.05).  We did not collect probiotics data for the healthy control group.

Comment 5. Did symptoms differ between individuals on probiotics and those without probiotics?

Reply. We did not compare the symptoms based on probiotics consumption, but this is a question for a future study as we plan to expand our research on the role of gut microbiome in FAPDs pediatric population. We plan to monitor the symptoms and the psychological stress as we collect stool samples from FAPDs patients over time.  

Comment 6. Did you assure that individuals were not on antibiotics in advance of every stool sample?

Reply. Antibiotics consumption is an exclusion factor from the study as outlined in the study methodology. As the samples were collected three times from every subject, the research coordinator made sure that no antibiotics were consumed before sending the stool sample.

Comment 7. Could you add some discussion on the function of taxae that were depicted with significant differences between groups?

Reply. The discussion was added in the manuscript as per reviewer request. We added the role of some bacterial taxa in SCFA production in the gut environment and their potential roles in the inflammation process. As we expand our future studies in FAPDs gut microbiome, fecal metabolites and functional genomics are underway to analyze the byproducts of gut microbiome between FAPDs and healthy control. Some of such bacterial taxa have probiotics effects as well.

Comment 8. Most authors use italic letters only for Genera and lower taxonomic levels - the use of italic letters also for phyla is a little confusing

Reply. The comment was addressed in the revised manuscript. We kept italics for genus level.

Reviewer 2 Report

The manuscript is well written. However, I failed to understand what was the criteria of sample collection. The authors mentioned the sample collection was done throughout the year? Was sample collection done once from each subject? Apart from Antibiotics, have subjects consumed any probiotics or prebiotics or any supplement in the last 5-6 months.  

Authors should provide an additional heatmap of dominant taxa at the genus level, as it is hard to understand the heatmap as presented. 

There should be a separate conclusion section highlighting the significance of the study and future directions.

Author Response

Reviewer 2

Comments. The manuscript is well written. However, I failed to understand what the criteria of sample collection was. The authors mentioned the sample collection was done throughout the year. Was sample collection done once from each subject? Apart from Antibiotics, have subjects consumed any probiotics or prebiotics or any supplement in the last 5-6 months.  Authors should provide an additional heatmap of dominant taxa at the genus level, as it is hard to understand the heatmap as presented. 

There should be a separate conclusion section highlighting the significance of the study and future directions.

Reply. Thanks for the comments about the manuscript. The sample criteria is as follows:

  1. Recruitment of FAPDs patients and healthy controls of the same age group range.
  2. Set up inclusion and exclusion criteria for the study.
  3. Stool samples were collected from every participants (controls and FAPDs) three times throughout the year.
  4. As Florida is absent of weather seasonality due to mild weather, we compared the gut microbiome in two timepoints (school versus summer vacation).

Antibiotics consumption was an exclusion criteria, but probiotics consumption was not due to the fact that some FAPDs patients consume probiotics as part of their Treatment As Usual (TAU). As we mentioned earlier, no significant differences between FAPDs gut microbiome as a result to the probiotics. We don’t have data regarding pre/probiotics consumption in the last 5-6 months.

We replaced the heatmap as per the request of the reviewer, and an additional heatmap was provided in the revised manuscript. Finally, the significance of the study and future direction are discussed in section 5.